# Aperture-Level Simultaneous Transmit and Receive Simplified Structure Based on Hybrid Beamforming of Switching Network

**Hongbin Yi, Xizhang Wei \*, Tairan Lin, Yanqun Tang, Mingcong Xie and Dujuan Hu** 

Electronics and Communication Engineering, Sun Yat-Sen University, Guangming District, Shenzhen 518107, China
\* Correspondence: weixzh7@mail.sysu.edu.cn

**Abstract:** With the increasing competition for spectrum resources, the technology of simultaneous transmit and receive (STAR) is attracting more and more attention. However, full digital aperture-level simultaneous transmit and receive (FD-ALSTAR) is difficult to implement in a large-scale array with high frequency and bandwidth due to its high hardware cost and high power consumption. Therefore, this paper combines FD-ALSTAR with hybrid beamforming technology and proposes two categories and four types of aperture-level simultaneous transmit and receive simplified structures based on hybrid beamforming to reduce the number of RF links (NRF), hardware cost, and operation power consumption. In view of the complexity of the hardware of the fully connected hybrid beamforming structure and the low amplitude and phase control accuracy of the partially connected hybrid beamforming structure, an aperture-level simultaneous transmit and receive simplified structure based on hybrid beamforming of switching network (HBF-SN-ALSTAR) is proposed, and the mathematical model is established. The simulation results show that the simplified structure proposed in this paper can effectively reduce the NRF and power consumption, increase system redundancy, and improve system reliability. In a $144 \times 144$ antenna array, under the condition that NRF = 16 of HBF-SN-ALSTAR, that is, 1/9 of the number of FD-ALSTAR RF links, the **effective isotropic isolation** (EII) of the system is only 17 dB less than that of the FD-ALSTAR. The experimental results fully prove the effectiveness of the simplified structure.

**Keywords:** aperture-level simultaneous transmit and receive; RF links; hybrid beamforming

## 1. Introduction

With the popularity of intelligent mobile terminal devices, the demand for spectrum is growing. STAR technology has been widely studied because it can effectively use spectrum resources. Compared with STAR technology, traditional frequency division duplex (FDD) and time division duplex (TDD) approaches require twice the frequency/time resources, while STAR technology can realize duplex technology at the same frequency and the same time, so it has a broader application prospect [1]. However, in modern communication and radar systems, antenna arrays are usually used to receive very weak RF signals. Because the configuration of the transceiver arrays is close, signal leakage from the transmitting end to the receiving end is very harmful to STAR technology. Therefore, signal leakage must be reduced or eliminated [2]. In order to solve this technical problem and realize STAR technology, relevant research mainly explores effective self-interference cancellation (SIC) methods from three fields of propagation domain: analog domain and digital domain, one field promoted separately, or a combination of several schemes [3].

The FD-ALSTAR structure is provided in [4]. As shown in Figure 1, it can realize ALSTAR technology in the full digital arrays. In Figure 1, $x$ is the desired signal to be transmitted, $s$ represents the external signal(s) of interest, $y$ is the received signal, and $y'$ removes self-interference from the received signals $y$, $n_t$, $n_r$, and $n_0$ represent transmitter noise, receiver noise and observation channel noise, respectively. $H_0$ represents the fixed

attenuator to prevent observation channel saturation, $M$ is the mutual coupling channel matrix between transmitting and receiving elements, and $b_t$, $b_r$, and $b_c$ are the transmit beamforming vector, receive beamforming vector, and multi-channel filter, respectively.

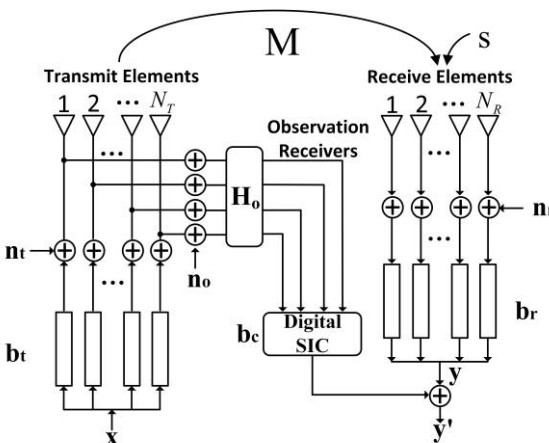

**Figure 1.** FD-ALSTAR structure.

In [5], through the FD-ALSTAR structure, using a digital transceiver on each antenna array element, dynamically allocating the transceiver units in the array, and only using digital beamforming and SIC technology, the authors showed that it is possible to replace the observation noise and transmission noise in the received signal, as well as eliminate the nonlinear components in the RF links, and obtain the digital beamforming vectors at the transmitter and receiver through iterative optimization algorithms. This makes it possible to realize FD-ALSTAR.

However, with the increase in array size and bandwidth, and the increase in the frequency, it is difficult to realize FD-ALSTAR. The main reason is that each antenna of a large-scale full digital array is equipped with a dedicated radio frequency (RF) link [6], and each RF link contains a data converter, mixer, power amplifier (PA), and Analog-to-Digital Converter/Digital-to-Analog Converter (ADC/DAC) [7], which is not only expensive but also has huge power consumption. For this reason, we propose the use of hybrid beamforming technology to reduce the NRF in the system [8], to make the application of ALSTAR technology realizable. At present, hybrid beamforming has two typical structures. One is a fully connected structure (FC), as shown in Figure 2a. Each RF link is connected to all antennas through analog phase shifters with the same number of antennas. The other is a partially connected structure (PC), as shown in Figure 2b. An RF link connects a fixed subarray, which can greatly reduce the system hardware complexity [9]. In Figure 2, $F_{BB}$ and $F_{RF}$ represent the digital baseband beamforming vector and analog RF beamforming vector of the transmitting end, respectively, and $N_S$, $N_T$, and $N_T^{RF}$ are the number of data streams, transmit antennas, and RF chains at the transmitting end, respectively.

For the FC structure of hybrid beamforming, the studied algorithms include the zero-forcing (ZF) algorithm [10], the orthogonal matching pursuit (OMP) algorithm [11], etc. For the PC structure of hybrid beamforming, relevant optimization algorithms include the dynamic sub-array algorithm [12], the water-filling algorithm [13], the iterative precoding and combining algorithm [14], etc. The authors of [15], based on the manifold optimization iterative algorithm, proposed the MO-AltMin algorithm for FC and the SDR-AltMin algorithm for PC. Both algorithms can approach the optimal spectral efficiency of full digital arrays in communication technology under the condition of fewer NRF.

For this reason, this paper combines hybrid beamforming technology with FD-ALSTAR and proposes two categories and four types of aperture-level simultaneous transmit and receive simplified structures based on hybrid beamforming (HBF-ALSTAR) according to the structure of the hybrid beamforming and the different positions of observation signals. Due to the special hardware configuration of the FC structure, it can make full use of the

degrees of freedom of beamforming provided by the limited RF links, improve the accuracy of the beamforming vector, and ensure the EII of the system. However, with the increase in RF links, the hardware structure will become complex, costly, and difficult to implement. Although PC structures can effectively reduce the hardware complexity of the system, because their RF links are connected to fixed sub-array, even if complex optimization algorithms are used, it is difficult to overcome their original hardware shortcomings and ensure the amplitude and phase accuracy of the beamforming vector, resulting in poor EII. In order to better balance the requirements of hardware complexity and EII, this paper uses the previous work in [16] for reference and adds a switching network in HBF-ALSTAR to achieve dynamic grouping of arrays to reduce the NRF and improve the EII.

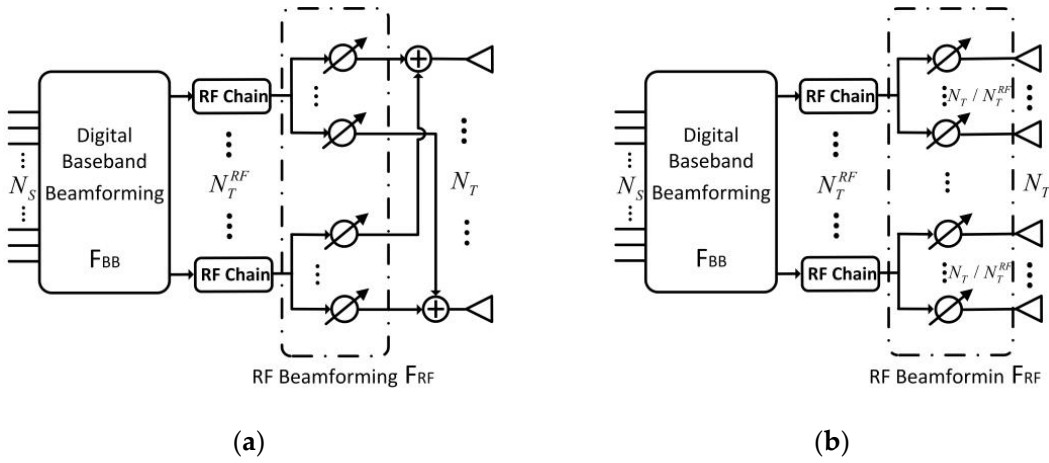

(**a**)                                    (**b**)

**Figure 2.** (**a**) Fully connected structure of hybrid beamforming; (**b**) Partially connected structure of hybrid beamforming.

The main contribution of this paper is to combine hybrid beamforming technology with FD-ALSTAR and propose four simplified structures of HBF-ALSTAR. In view of the high hardware complexity of the FC structure and the low amplitude and phase control accuracy, and the poor EII of the PC structure, a simplified structure of HBF-ALSTAR based on a switching network is proposed, which can reduce the NRF, power consumption, and cost, and increase system redundancy and reliability under the condition of ensuring system EII, beam pattern, and gain. This approach makes ALSTAR technology more conducive to engineering implementation. Table 1 lists the acronyms used in this paper.

**Table 1.** Table of Acronym.

| Acronym | Definition |
| --- | --- |
| STAR | Simultaneous Transmit and Receive |
| FD-ALSTAR | Full Digital Aperture-Level Simultaneous Transmit and Receive |
| NRF | Number of RF Links |
| HBF-SN-ALSTAR | Aperture-Level Simultaneous Transmit and Receive Simplified Structure Based on Hybrid Beamforming of Switching Network |
| EII | Effective Isotropic Isolation |
| FDD | Frequency Division Duplex |
| TDD | Time Division Duplex |
| SIC | Self-Interference Cancellation |
| ADC/DAC | Analog-to-Digital Converter /Digital-to-Analog Converter |
| FC | Fully Connected |
| PC | Partially Connected |
| ZF | Zero Forcing Algorithm |
| OMP | Orthogonal Matching Pursuit Algorithm |
| MO-AltMin | Manifold Optimization Based Hybrid Precoding for the Fully connected |

**Table 1.** *Cont.*

| Acronym | Definition |
| --- | --- |
| SDR-AltMin | Semidefinite Relaxation Based Hybrid Precoding for the Partially connected |
| HBF-ALSTAR | Aperture-Level Simultaneous Transmit and Receive Simplified Structure Based on Hybrid Beamforming |
| HBF-FC-ALSTAR-AO | Aperture-Level Simultaneous Transmit and Receive Simplified Structure Based on Antenna End Observation of Fully Connected Hybrid Beamforming |
| HBF-PC-ALSTAR-AO | Aperture-Level Simultaneous Transmit and Receive Simplified Structure Based on Antenna End Observation of Partially Connected Hybrid Beamforming |
| HBF-FC-ALSTAR-RFO | Aperture-Level Simultaneous Transmit and Receive Simplified Structure Based on RF Links Observation of Fully Connected Hybrid Beamforming |
| HBF-PC-ALSTAR-RFO | Aperture-Level Simultaneous Transmit and Receive Simplified Structure Based on RF Links Observation of Partially Connected Hybrid Beamforming |
| HBF-SN-ALSTAR-AO | Aperture-Level Simultaneous Transmit and Receive Simplified Structure Based on antenna end observation of Hybrid Beamforming of Switching Network |
| HBF-SN-ALSTAR-RFO | Aperture-Level Simultaneous Transmit and Receive Simplified Structure Based on RF links observation of Hybrid Beamforming of Switching Network |
| EIRP | Effective Isotropic Radiated Power |
| EIS | Effective Isotropic Sensitivity |
| HBF-SN-H-ALSTAR | Aperture-Level Simultaneous Transmit and Receive Simplified Structure Based on Hybrid Beamforming of Switching Network of Statistical Dynamic Grouping |
| HBF-SN-A-ALSTAR | Aperture-Level Simultaneous Transmit and Receive Simplified Structure Based on Hybrid Beamforming of Switching Network of Average Dynamic Grouping |
| AOD | Angles of Departure |
| AOA | Angles of Arrival |
| USPA | Uniform Square Array |

## 2. System Model

### 2.1. Aperture-Level Simultaneous Transmit and Receive Simplified Structure Based on Hybrid Beamforming (HBF-ALSTAR)

The FC and PC structures of hybrid beamforming are combined with FD-ALSTAR, and according to the location of the observation signals, four different types of simplified structures are obtained, namely, the aperture-level simultaneous transmit and receive simplified structure based on antenna end observation of fully connected hybrid beamforming (HBF-FC-ALSTAR-AO); the aperture-level simultaneous transmit and receive simplified structure based on antenna end observation of partially connected hybrid beamforming (HBF-PC-ALSTAR-AO); the aperture-level simultaneous transmit and receive simplified structure based on RF links observation of fully connected hybrid beamforming (HBF-FC-ALSTAR-RFO); and the aperture-level simultaneous transmit and receive simplified structure based on RF links observation of partially connected hybrid beamforming (HBF-PC-ALSTAR-RFO). These structures are shown in Figure 3a–d. In Figure 3, $x$, $y$, $y'$, $M$, $n_t$, $n_r$, and $n_o$ are the same as in Figure 1. $F_{BB}$ and $F_{RF}$ are the digital baseband beamforming vector and the analog RF beamforming vector of the transmitting end, respectively, $W_{BB}$ and $W_{RF}$ are the digital baseband beamforming and the analog RF beamforming of the receiving end, respectively, and $G_{BB}$ represents a multi-channel filter to achieve SIC.

In the two simplified structures shown in Figure 3a,b, the signal model and principle of the system are largely similar to that of FD-ALSTAR. The main difference between the two structures is that in Figure 3a, the FC structure of hybrid beamforming, and in Figure 3b,

the PC structure of hybrid beamforming replaces the antenna arrays of the transmitter and receiver of FD-ALSTAR respectively.

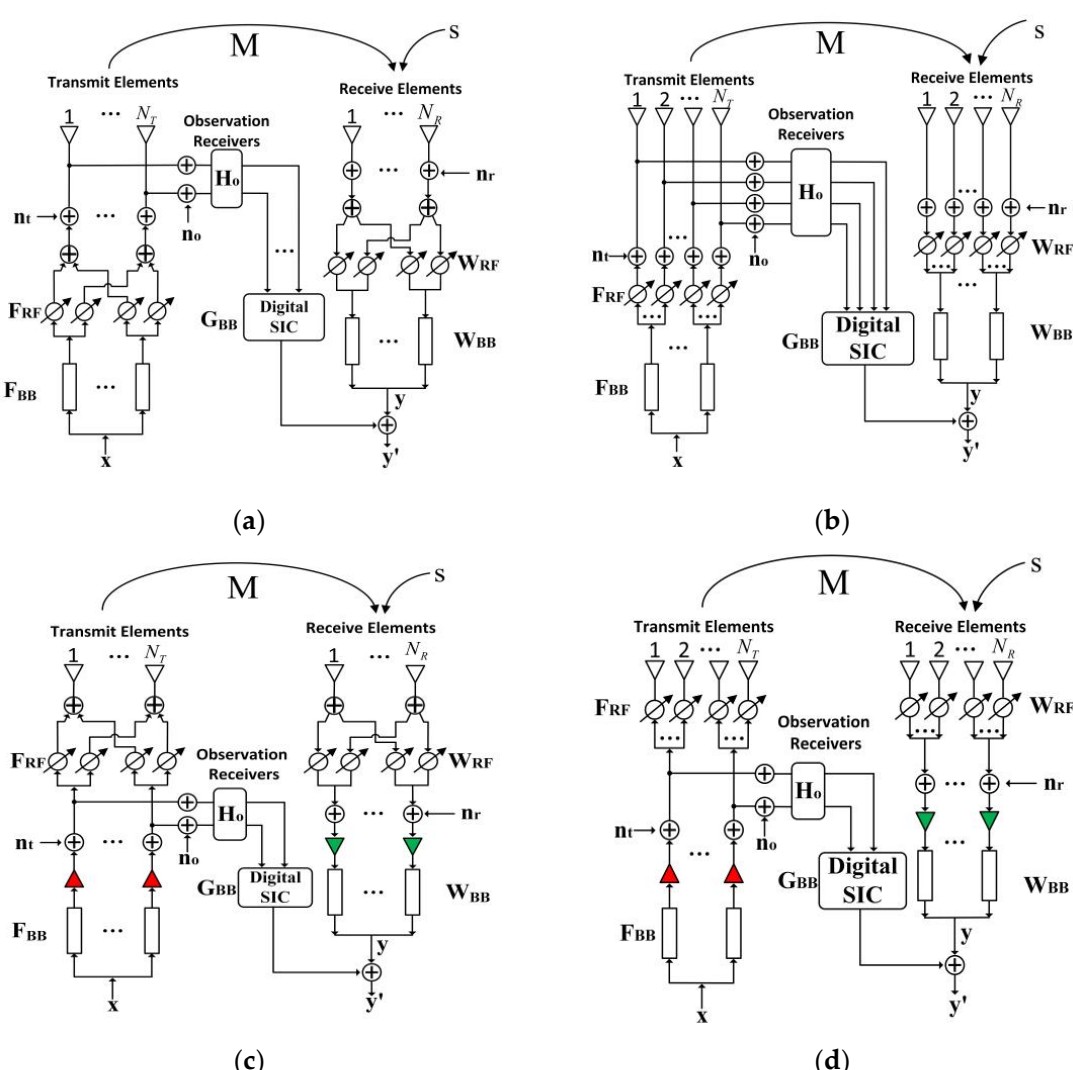

**Figure 3.** (**a**) HBF-FC-ALSTAR-AO; (**b**) HBF-PC-ALSTAR-AO; (**c**) HBF-FC-ALSTAR-RFO; (**d**) HBF-PC-ALSTAR-RFO.

The difference between Figure 3a–d is that the observation signal is selected from the RF links in Figure 3c,d. The main reason for the design of these two simplified structures is that the power level noise and nonlinear components in the system are mainly from the RF links, which can, thus, be observed effectively in Figure 3c,d. This can not only reduce the NRF at the transmitter and receiver but also greatly reduce the NRF in the observation links. The hardware structure of the whole system is simpler and the digital signal processing pressure is further reduced. However, under these two structures, the observation link cannot easily observe the amplitude and phase errors of the RF links after the sampling points. For this reason, in the structure of Figure 3c,d, the amplitude and phase correction unit is required to realize an ALSTAR approach.

*2.2. Aperture-Level Simultaneous Transmit and Receive Simplified Structure Based on Hybrid Beamforming of Switching Network (HBF-SN-ALSTAR)*

In order to further improve the EII of the HBF-PC-ALSTAR simplified structure and overcome the disadvantage of RF links connecting fixed antenna subarray, a switching network is introduced in the simplified structure to realize the dynamic link between RF

links and antenna subarray. The two structures are, respectively: the HBF-SN-ALSTAR based on antenna end observation (HBF-SN-ALSTAR-AO) and the HBF-SN-ALSTAR based on RF links observation (HBF-SN-ALSTAR-RFO), as shown in Figure 4a,b.

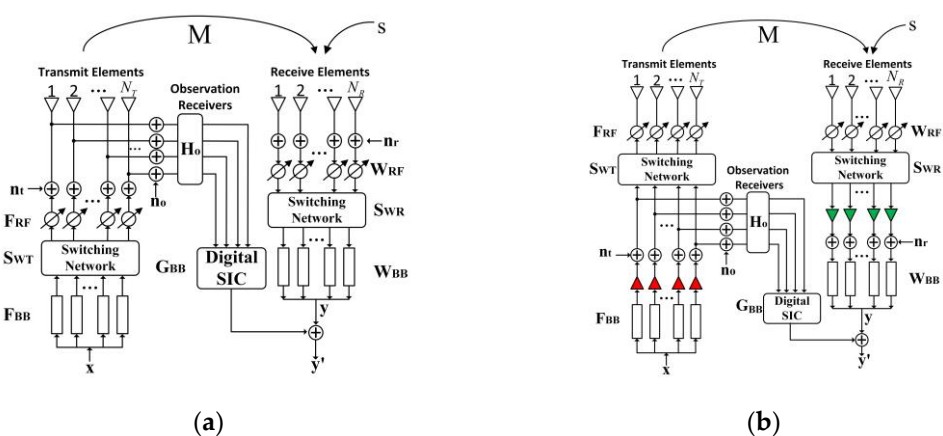

**(a)** **(b)**

**Figure 4.** (**a**) HBF-SN-ALSTAR-AO; (**b**) HBF-SN-ALSTAR-RFO.

The basic mathematical model derivation is based on the literature [5]. The mathematical models of the structure in Figures 3 and 4 have differences only in the dimensions of parameters, and there are no obvious other differences. Therefore, the derivation of the signal mathematical models of the six simplified structures in this paper is mainly based on the HBF-SN-ALSTAR-AO, and the signal models of the other simplified structures will not be derived separately.

The mathematical model of HBF-SN-ALSTAR-AO is based on [14]. The transmitted signal vector is $t(n) \in \mathbb{C}^{N_T \times 1}$, where $n$ is the time index and:

$$t(n) = F_{RF}S_{WT}F_{BB}x(n) + n_t(n) \tag{1}$$

where $x(n) \in \mathbb{C}^{1 \times 1}$ denotes the desired signal to be transmitted with $\mathrm{E}\left[|x(n)|^2\right] = 1$, $F_{BB} \in \mathbb{C}^{N_T^{RF} \times 1}$ is the digital baseband beamforming vector at the transmitter, $S_{WT} \in \mathbb{C}^{N_T \times N_T^{RF}}$ is the transmit switching network dynamic allocation matrix, and $F_{RF} \in \mathbb{C}^{N_T \times N_T}$ is the analog beamforming matrix at the transmitter, where $F_{RF}$ is the diagonal matrix of which the number of non-zero elements is $N_T$ and all non-zero elements in $F_{RF}$ should satisfy the unit modulus constraints, i.e., $\left|(F_{RF})_{i,j}\right| = 1$. The number of non-zero elements in $S_{WT}$ is also $N_T$, and they are all equal to 1. $n_t(n) \in \mathbb{C}^{N_T \times 1} \sim N(0, \sigma_t^2)$ is zero-mean, complex additive white Gaussian noise (AWGN) transmitted noise with the covariance matrix $N_t = \mathrm{E}\left[n_t n_t^H\right] = Diag\left((F_{RF}S_{WT}F_{BB})(F_{RF}S_{WT}F_{BB})^H\right)/\eta_t$, where $\eta_t$ is the signal-to-noise ratio (SNR) of each transmitter. The received signal vector $r(n) \in \mathbb{C}^{N_R \times 1}$ is expressed as follows:

$$r(n) = M(F_{RF}S_{WT}F_{BB}x(n) + n_t(n)) + s(n) \tag{2}$$

where $s(n) \in \mathbb{C}^{N_R \times 1}$ represents the external signal(s) of interest, with $R_{ss} = \mathrm{E}\left[ss^H\right]$, $M \in \mathbb{C}^{N_R \times N_T}$ denotes the mutual coupling channel between transmitter and receiver. The received signals are transformed by receive beamforming as follows:

$$y(n) = (W_{RF}S_{WR}W_{BB})^H(M(F_{RF}S_{WT}F_{BB}x(n) + n_t(n)) + s(n) + n_r(n)) \tag{3}$$

where $W_{BB} \in \mathbb{C}^{N_R^{RF} \times 1}$ is the digital baseband beamforming vector at the receiver, $S_{WR} \in \mathbb{C}^{N_R \times N_R^{RF}}$ is the receive switching network dynamic allocation matrix, and $W_{RF} \in \mathbb{C}^{N_R \times N_R}$ is the analog beamforming matrix at the receiver, where $W_{RF}$ is the diagonal matrix of which the number of non-zero elements is $N_R$ and all non-zero elements in $W_{RF}$ should

satisfy the unit modulus constraints, i.e., $\left|(W_{RF})_{i,j}\right| = 1$ for nonzero elements. $n_r(n) \sim N(0, \sigma_r^2)$ expresses zero-mean complex Gaussian receiver noise with $N_r = \mathrm{E}\left[n_r n_r^H\right] = Diag\left(\mathrm{E}\left[rr^H\right]\right)/\eta_r + \sigma_r^2 I$. $\eta_r$ denotes the SNR of each receiver. and $\sigma_r^2$ is the receiver thermal noise power.

The resulting SINR of the received beam is expressed as:

$$\mathrm{SINR} = \frac{P_y^s}{P_y^x + P_y^{nt} + P_y^{nr}} \tag{4}$$

where the power of the external signal(s) of interest in the received signal $y(n)$ is:

$$P_y^s = (W_{RF} S_{WR} W_{BB})^H R_{SS} (W_{RF} S_{WR} W_{BB}) \tag{5}$$

the power of the transmitter signal coupled to the received signal $y(n)$ is:

$$P_y^x = (W_{RF} S_{WR} W_{BB})^H M (F_{RF} S_{WT} F_{BB})(F_{RF} S_{WT} F_{BB})^H M^H (W_{RF} S_{WR} W_{BB}) \tag{6}$$

the power of the noise at the transmitter coupled to the received signal $y(n)$ is:

$$P_y^{nt} = (W_{RF} S_{WR} W_{BB})^H M N_t M^H (W_{RF} S_{WR} W_{BB}) \tag{7}$$

the power of receiver noise in the received signal $y(n)$ is:

$$P_y^{nr} = (W_{RF} S_{WR} W_{BB})^H N_r (W_{RF} S_{WR} W_{BB}) \tag{8}$$

and the observed signal vector $o(n) \in \mathbb{C}^{N_T \times 1}$ is:

$$o(n) = H_o(F_{RF} S_{WT} F_{BB} x(n) + n_t(n) + n_o(n)) \tag{9}$$

where $H_o \in \mathbb{C}^{N_T \times N_T}$ is a diagonal matrix which represents the fixed attenuator between each transmit channel and its corresponding observation receive channel, to prevent observation channel saturation. The observation channel is AWGN with $n_o(n) \sim N(0, \sigma_o^2)$ where $N_o = \mathrm{E}\left[n_o n_o^H\right] = Diag\left((F_{RF} S_{WT} F_{BB})(F_{RF} S_{WT} F_{BB})^H\right)/\eta_r$. The observed signal passed through the digital SIC $G_{BB}$ and added to the signal $y(n)$ is:

$$y'(n) = y(n) + G_{BB}^H o(n) \tag{10}$$

setting $G_{BB}^H = -(W_{RF} S_{WR} W_{BB})^H M H_o^{-1}$, the final signal is:

$$y' = (W_{RF} S_{WR} W_{BB})^H (s(n) + n_r(n) - M n_o(n)) \tag{11}$$

we can see that the transmit noise $n_t(n)$ has been replaced with the observation noise $n_o(n)$, and the observation noise power is:

$$P_y^{no} = (W_{RF} S_{WR} W_{BB})^H M N_o M^H (W_{RF} S_{WR} W_{BB}) \tag{12}$$

The amount of residual noise $P_y^{no}$ and $P_y^{nr}$ will still reduce the EII. The correlation matrix of residual noise can be expressed as:

$$
\begin{aligned}
N_r &= \eta_r^{-1} Diag(R_{ss}) + \eta_r^{-1} Diag\left(M(F_{RF} S_{WT} F_{BB})(F_{RF} S_{WT} F_{BB})^H M^H\right) \\
&\quad + \eta_r^{-1} \eta_t^{-1} Diag\left(M Diag\left((F_{RF} S_{WT} F_{BB})(F_{RF} S_{WT} F_{BB})^H\right) M^H\right) + \sigma_r^2 I
\end{aligned} \tag{13}
$$

which can be approximated as:

$$N_r \approx \eta_r^{-1} Diag\left(M(F_{RF}S_{WT}F_{BB})(F_{RF}S_{WT}F_{BB})^H M^H\right)$$
$$+\eta_r^{-1}\eta_t^{-1} Diag\left(MDiag\left((F_{RF}S_{WT}F_{BB})(F_{RF}S_{WT}F_{BB})^H\right)M^H\right) + \sigma_r^2 I \tag{14}$$

The total residual noise in the receive beam can be expressed as a quadratic form of the receive beamformer $W_{RF}S_{WR}W_{BB}$:

$$P_n = (W_{RF}S_{WR}W_{BB})^H M_{br}(W_{RF}S_{WR}W_{BB}) \tag{15}$$

where:

$$M_{br} = \eta_r^{-1} MDiag\left((F_{RF}S_{WT}F_{BB})(F_{RF}S_{WT}F_{BB})^H\right)M^H$$
$$+\eta_r^{-1} Diag\left(M(F_{RF}S_{WT}F_{BB})(F_{RF}S_{WT}F_{BB})^H M^H\right)$$
$$+\eta_r^{-1}\eta_t^{-1} Diag\left(MDiag\left((F_{RF}S_{WT}F_{BB})(F_{RF}S_{WT}F_{BB})^H\right)M^H\right)$$
$$+\sigma_r^2 I \tag{16}$$

According to the fact that $a^H Diag(bb^H)a = b^H Diag(aa^H)b$, the formula (15) can be written as:

$$P_n = (F_{RF}S_{WT}F_{BB})^H M_{bt}(F_{RF}S_{WT}F_{BB}) \tag{17}$$

where:

$$M_{bt} = \eta_r^{-1} Diag\left(M^H(W_{RF}S_{WR}W_{BB})(W_{RF}S_{WR}W_{BB})^H M\right)$$
$$+\eta_r^{-1} M^H Diag\left((W_{RF}S_{WR}W_{BB})(W_{RF}S_{WR}W_{BB})^H\right)M$$
$$+\eta_r^{-1}\eta_t^{-1} Diag\left(MDiag\left((W_{RF}S_{WR}W_{BB})(W_{RF}S_{WR}W_{BB})^H\right)M^H\right)$$
$$+\frac{\sigma_r^2}{P_T} I \tag{18}$$

and meet the system's transceiver power limit, respectively:

$$\|F_{RF}S_{WT}F_{BB}\|^2 = P_T, \|W_{RF}S_{WR}W_{BB}\|^2 = 1 \tag{19}$$

Accordingly, the EII metric, which is the ratio of Effective Isotropic Radiated Power (EIRP) to Effective Isotropic Sensitivity (EIS):

$$EII = \frac{EIRP}{EIS} \tag{20}$$

EIRP—the power required by a theoretical isotropic transmitter to provide equivalent illumination in the desired direction—can be defined as:

$$EIRP(\phi, \theta, F_{RF}F_{BB}) = g(\phi, \theta)(F_{RF}S_{WT}F_{BB})^H q_t(\phi, \theta)q_t(\phi, \theta)^H (F_{RF}S_{WT}F_{BB}) \tag{21}$$

where:

$$q_t(\phi, \theta) = e^{-j\frac{2\pi}{\lambda}(X_t \cos(\phi)\sin(\theta) + Y_t \sin(\phi)\sin(\theta))} \tag{22}$$

EIS—the noise floor of a theoretical isotropic receiver with an equivalent sensitivity in the desired direction—can be defined as:

$$EIS = \frac{P_n}{G_r} \tag{23}$$

where:

$$G_r(\phi, \theta, W_{RF}S_{WR}W_{BB}) = g(\phi, \theta)(W_{RF}S_{WR}W_{BB})^H q_r(\phi, \theta)q_r(\phi, \theta)^H (W_{RF}S_{WR}W_{BB}) \tag{24}$$

Therefore, the Formula (20) can be written as follows:

$$\begin{aligned}
\text{EII} &= G_r(\phi,\theta, W_{RF}S_{WR}W_{BB})g(\phi,\theta)\frac{(F_{RF}S_{WT}F_{BB})^H q_t(\phi,\theta)q_t^H(\phi,\theta)(F_{RF}S_{WT}F_{BB})}{(F_{RF}S_{WT}F_{BB})^H M_{bt}(F_{RF}S_{WT}F_{BB})}\\
&= \text{EIRP}(\phi,\theta, F_{RF}S_{WT}F_{BB})g(\phi,\theta)\frac{(W_{RF}S_{WR}W_{BB})^H q_r(\phi,\theta)q_r^H(\phi,\theta)(W_{RF}S_{WR}W_{BB})}{(W_{RF}S_{WR}W_{BB})^H M_{bt}(W_{RF}S_{WR}W_{BB})}
\end{aligned} \tag{25}$$

After obtaining the transmit beamforming vector $F_{RF}S_{WT}F_{BB}$ and the receive beamforming vector $W_{RF}S_{WR}W_{BB}$ through the alternate iterative optimization algorithm, we can calculate the corresponding $F_{RF}$, $S_{WT}$, $F_{BB}$ and $W_{RF}$, $S_{WR}$, $W_{BB}$, respectively.

Taking the transmitter as an example, the main principle is as follows:

Through the switching network, each RF link is dynamically connected with the antennas. The main method is to group according to the amplitude values of the transmit and receive beamforming vectors iteratively optimized in [5]. The antennas of each group are dynamically connected to an RF link through the switching network—that is, the RF link provides the amplitude of the beamforming vector of the connected subarray. In Figure 5, the green switching network node indicates the on state, and the black switching network node indicates the off state.

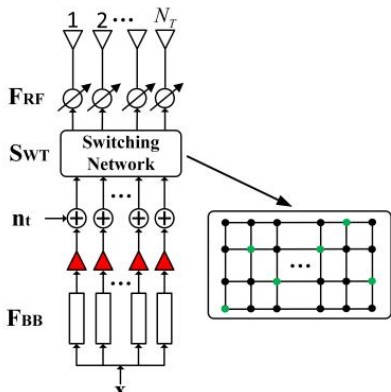

**Figure 5.** HBF-SN-LASTAR local structure diagram of the transmitter.

In this paper, when grouping the amplitude of the target beamforming vector, three dynamic grouping methods are explored:

1. Grouping according to the amplitude of the optimal transmit beamforming vector $F_{RF}S_{WT}F_{BB}$, which is called optimal dynamic grouping (HBF-SN-ALSTAR);

2. Grouping according to the statistical mean values of the amplitudes of the $F_{RF}S_{WT}F_{BB}$ corresponding to the $M$ parameters of different channels, which is called statistical dynamic grouping (HBF-SN-H-ALSTAR);

3. The fixed number of antennas are connected to each RF chain, which is similar to the PC structure, but the difference is that the array elements with similar amplitudes are dynamically combined, which is called average dynamic grouping (HBF-SN-A-ALSTAR).

Considering grouping methods 2 and 3, statistical dynamic grouping and average dynamic grouping are grouping methods with a fixed number of array elements, which can fix the working range of RF devices and further simplify system complexity and reduce hardware cost.

The mathematical model is derived as follows, setting $F_{opt} = F_{RF}S_{WT}F_{BB}$, and the objective function is:

$$\begin{aligned}
&\min && \left\| F_{opt} - F_{RF}S_{WT}F_{BB} \right\|_F\\
&s.t. && \left| (F_{RF})_{i,j} \right| = 1, \left\| F_{RF}S_{WT}F_{BB} \right\|^2 = P_t
\end{aligned} \tag{26}$$

Considering that analog beamforming is of infinite resolution or using high-resolution phase shifters, the phase value in the transmit beamforming vector $F_{opt}$ can be one-to-one

corresponding to the phase value in $F_{RF}$—that is, it can be completely consistent with the phase value, so only the amplitude information of the beamforming vector needs to be optimized in this objective function. Setting $F_{opt} = F_{RF}S_{WT}F_{BB}$, according to the previous amplitude grouping, to represent the number of antennas of the connected subarray of the RF link, then, in Formula (26), the best amplitude substitution is calculated as follows:

$$\min \sum_{i=1}^{N_T^{RF}} \sum_{j=1}^{n_i} \left| \left| F_{opt(j)} \right| - |F_i| \right|^2 (n_1 + n_2 + \cdots + n_{N_T^{RF}} = N_T) \tag{27}$$

The optimal amplitude of the RF link is:

$$|F_i| = \frac{\left( |F_{opt(1)}| + |F_{opt(2)}| + \cdots + |F_{opt(n_i)}| \right)}{n_i} \tag{28}$$

After determining the dynamic grouping matrix of the switching network $S_{WT}$ according to $|F_{opt}|$ divided by $N_T^{RF}$ radio frequency links, the transmit analog beamforming matrix is:

$$F_{RF} = Diag\left( F_{opt} / |F_{opt}| \right) \tag{29}$$

## 3. Simulation Experiment

### 3.1. Channel Parameter

In this paper, we chose the Saleh–Valenzuela mode [17]. The channel matrix $M$ is expressed as:

$$M = \sqrt{\frac{N_t N_r}{N_{cl} N_{ray}}} \sum_{i=1}^{N_{cl}} \sum_{l=1}^{N_{ray}} \alpha_{il} a_r(\phi_{il}^r, \theta_{il}^r) a_t(\phi_{il}^t, \theta_{il}^t)^H \tag{30}$$

where $N_{cl}$ and $N_{ray}$ represent the number of clusters and the number of rays in each cluster, respectively, and $\alpha_{il}$ denotes the gain of the $l$th ray in the $i$th propagation cluster. $\alpha_{il}$ are random variables following the complex Gaussian distribution $N(0, \sigma_{\alpha,i}^2)$, and $\sum_{i=1}^{N_{cl}} \sigma_{\alpha,i}^2 = \hat{\gamma}$ is the normalization factor to satisfy $E\left[ \|H\|_F^2 \right] = N_t N_r$. In addition, $a_r(\phi_{il}^r, \theta_{il}^r)$ and $a_t(\phi_{il}^t, \theta_{il}^t)$ represent the receive and transmit array response vectors, respectively, where $\phi_{il}^r(\phi_{il}^r)$ and $\theta_{il}^r(\theta_{il}^t)$ stand for azimuth and elevation angles of arrival and departure (AOAs and AODs), respectively. In this paper, we consider a uniform square planar array (USPA) with $\sqrt{N} \times \sqrt{N}$ antenna elements. Therefore, the array response vector corresponding to the $l$th ray in the $i$th cluster can be written as:

$$a(\phi_{il}, \theta_{il}) = 1/\sqrt{N}(1, \ldots, e^{j\frac{2\pi}{\lambda}d(p \sin\phi_{il} \sin\theta_{il} + q \cos\theta_{il})}, \ldots, e^{j\frac{2\pi}{\lambda}d((\sqrt{N}-1)\sin\phi_{il}\sin\theta_{il} + (\sqrt{N}-1)\cos\theta_{il})})^T \tag{31}$$

where $d$ and $\lambda$ are the antenna spacing and the signal wavelength, respectively, and $0 \leq p \leq \sqrt{N}$ and $0 \leq q \leq \sqrt{N}$ are the antenna indices in the 2D plane, respectively. Our structures and precoder algorithm can be used for more general models. We assume that perfect channel information can be obtained.

### 3.2. Experimental Parameters

In the experiment, a $12 \times 24$ antenna array is selected, i.e., $N_T = 144$, $N_R = 144$, the signal frequency is 30 GHz, the antenna spacing is $\lambda/2 = 5$ mm, the antenna gain is $g(\phi, \theta) = \pi \cos(\theta)$, the maximum scanning angle of the array is $60°$, the dynamic range of the transmitting channel is $\eta_t = 45$ dB, the dynamic range of the receiving channel is $\eta_r = 70$ dB, the thermal noise power of the receiving channel is $\sigma_r^2 = -91$ dB, and the noise coefficient is 3 dB. The transmit power is $P_T = 2500$ W. The channel parameters are set as $N_{cl} = 5$, $N_{ray} = 10$, the power of the respective cluster is $\sigma_{\alpha,i}^2 = 1$, and the azimuth and pitch angles of AOD and AOA follow the Laplacian distribution $[0, 2\pi]$, with the distribution

angle extending to 10°. All simulation results are achieved in 1000-channel data. In this paper, we select the MO-AltMin optimization algorithm of FC structures described in [15] and the SDR-AltMin optimization algorithm of PC structure for comparative experiments.

Figure 6a–d compare the system EII of the three categories of simplified structures under different RF link conditions. The EII of the HBF-PC-ALSTAR and the HBF-SN-ALSTAR increases with the increase in the number of RF links. In the experimental result, it can be seen that the EII of the HBF-FC-ALSTAR will fluctuate in performance, but on the whole, the HBF-FC-ALSTAR maintains a high isolation level. Compared with the HBF-PC-ALSTAR, the EII of the HBF-SN-ALSTAR has increased significantly. It can be seen from its structure that the increase in the switching network improves the control accuracy of the amplitude and phase, realizes the dynamic connectivity of the antennas, and increases the redundancy to a degree. When some of the RF links do not work, the rest of the RF links can still operate through the switching network, enhancing the stability and reliability of the system. Table 2 lists the number of different electronic devices under the condition of NRF = 24 for three simplified structures.

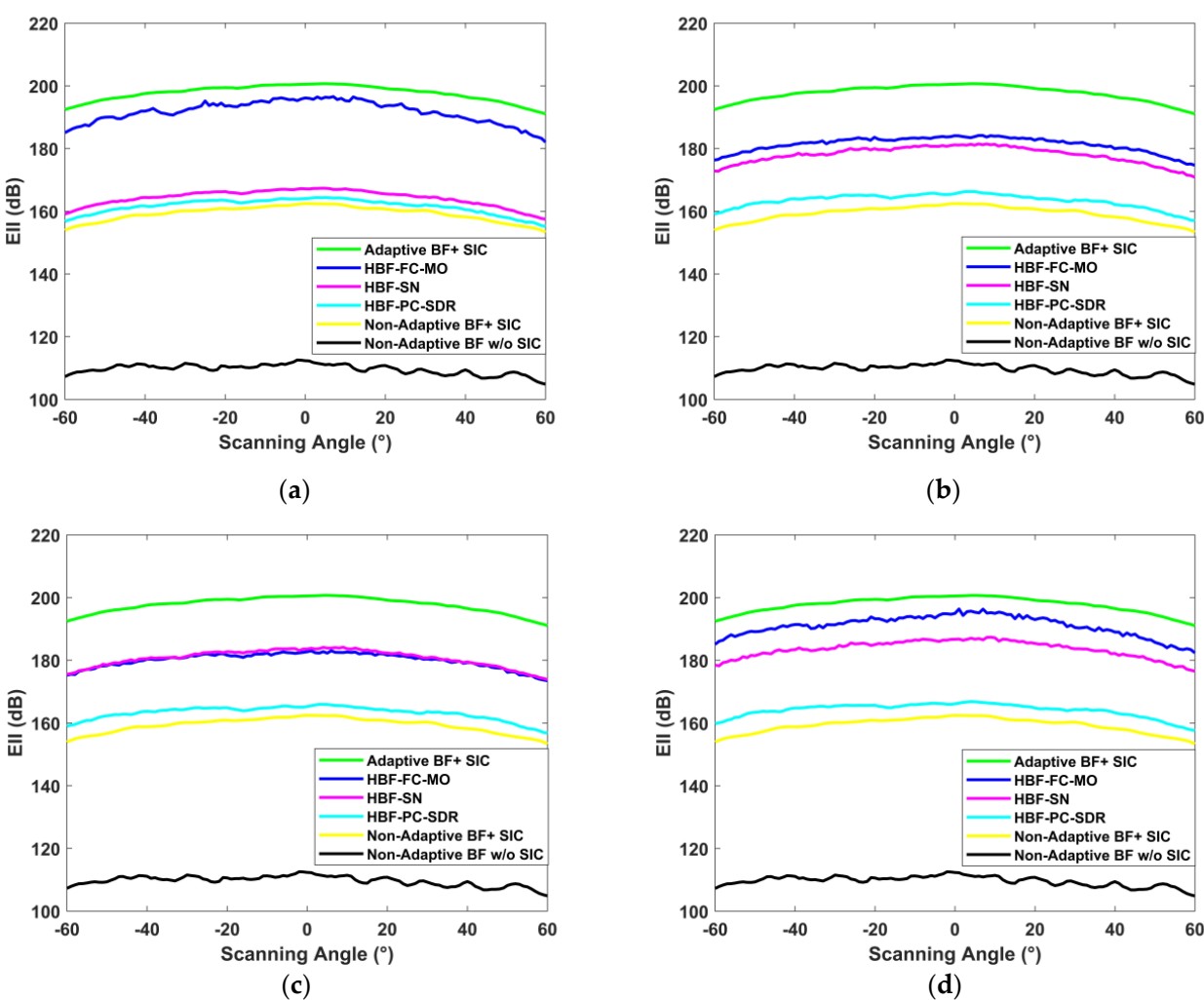

**Figure 6.** (**a**) $NRF = 2$; (**b**) $NRF = 12$; (**c**) $NRF = 16$; (**d**) $NRF = 24$.

For the three different dynamic grouping structures shown in Figure 7, the first, HBF-SN-ALSTAR, uses direct dynamic grouping to achieve the best EII. The second is HBF-SN-H-ALSTAR which is slightly worse than the EII achieved by the first grouping method, but most of the scanning angles are better than the EII of the HBF-SN-A-ALSTAR. Relatively speaking, the HBF-SN-A-ALSTAR dynamic grouping method can also maintain a high EII level. Compared with the HBF-SN-ALSTAR and the HBF-SN-H-ALSTAR, the

HBF-SN-A-ALSTAR dynamic grouping method can further simplify the working range of RF devices such as power divider, power synthesizer, and PA in the hardware structure, and is more stable than the EII achieved by the HBF-SN-H-ALSTAR dynamic grouping method.

**Table 2.** The number of different electronic devices for three simplified structures. (NRF = 24).

| Electronic Device | HBF-FC-ALSTAR-RFO | HBF-PC-ALSTAR-RFO | HBF-SN-ALSTAR-A-RFO |
|---|---|---|---|
| DAC | 24 | 24 | 24 |
| ADC | 24 | 24 | 24 |
| PA | 24 | 24 | 24 |
| Frequency Mixer | 144 | 144 | 144 |
| Phase Shifter | 3456 | 144 | 144 |
| Switching Network | 0 | 0 | 1 (144 × 24) |
| Power Combiner | 144 (144 to 1) | 0 | 0 |
| Power Divider | 24 (1 to 144) | 24 (1 to 6) | 24 (1 to 6) |

* 144 × 24 represents the size of the switching network.

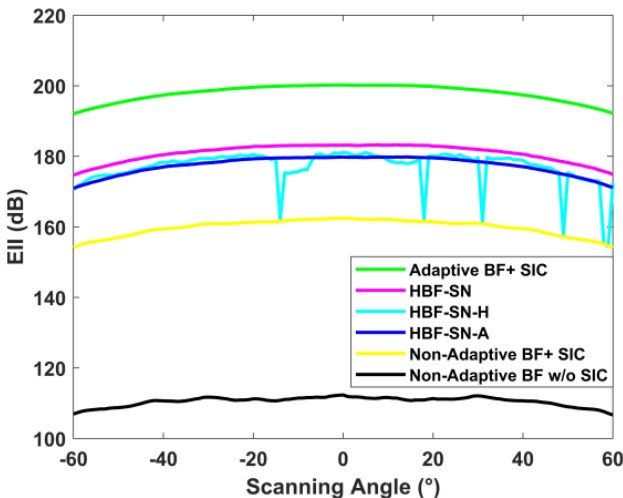

**Figure 7.** Comparison of three grouping methods of HBF-SN-ALSTAR ($NRF = 16$).

As can be seen in Figure 8, the EII difference between the HBF-PC-ALSTAR and the HBF-SN-ALSTAR and FD-ALSTAR decreases with the increase in the number of RF links. However, the EII difference between the HBF-PC-ALSTAR and FD-ALSTAR structures is large, more than 30 dB, and the performance improvement is not significant. The HBF-FC-ALSTAR reaches the local optimal solution at NRF = 2 but its performance will decline for NRF = 3–16 to some extent. When NRF > 16, the system performance will gradually approach that of the FD-ALSTAR with the increase in NRF, but its EII largely remains at the optimal level of the three types of simplified structures. In HBF-SN-ALSTAR, the EII difference decreases rapidly with the increase in NRF, especially when NRF = 16–17, when the EII performance achieved is better than the MO-AltMin algorithm of HBF-FC-ALSTAR. However, in the MO-AltMin algorithm selected for the FC structure in this paper, because its nested-loop structure contains the Kronecker product, which requires complex matrix calculation, the convergence speed is slow [18]. Although the HBF-PC-ALSTAR has a high degree of hardware simplification, because an RF link is connected to a fixed antenna subarray, it will bring a large system performance error. Especially in this structure, to achieve ALSTAR, the accuracy of the amplitude and phase of the beamforming vector is required to be high. In contrast, the algorithm corresponding to HBF-SN-ALSTAR is relatively simple, and for large-scale antenna arrays, the number of RF links saved is more significant.

Figures 9 and 10 show the beamforming vector pattern of the receiver at 0° and the array gain $G_t \times G_r$ of the three categories of simplified structure and the FD-ALSTAR, respectively. From the result, we can see that the array gain and the beamforming pattern

of the receiver are largely consistent with FD-ALSTAR, and there is no large main beam pointing deviation, sidelobe lifting, and gain change.

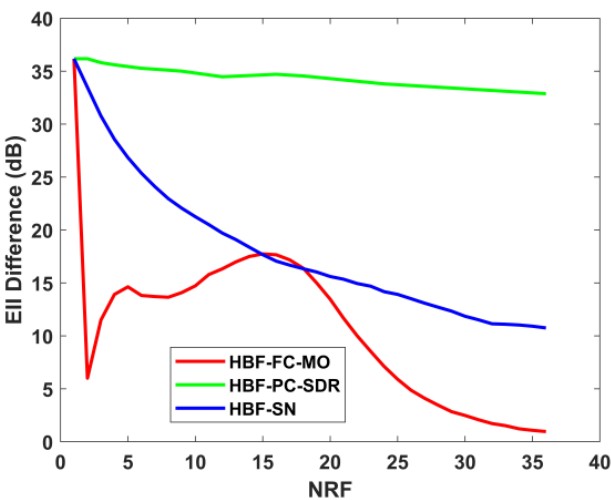

**Figure 8.** EII of three categories of simplified structures and FD-ALSTAR under different NRF conditions.

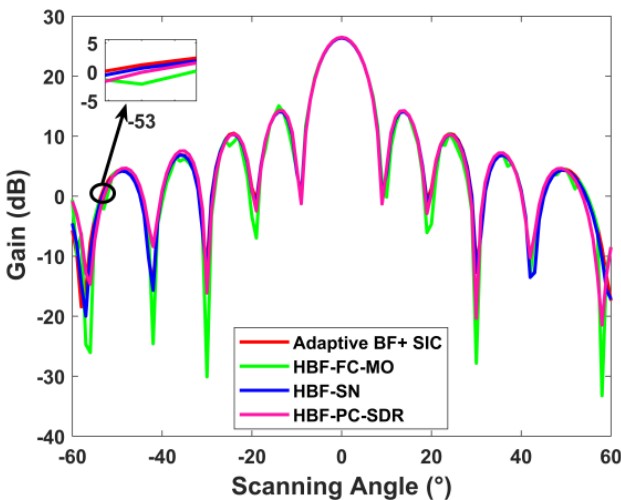

**Figure 9.** Pattern at $0°$ of three categories of simplified structure and FD-ALSTAR receiver.

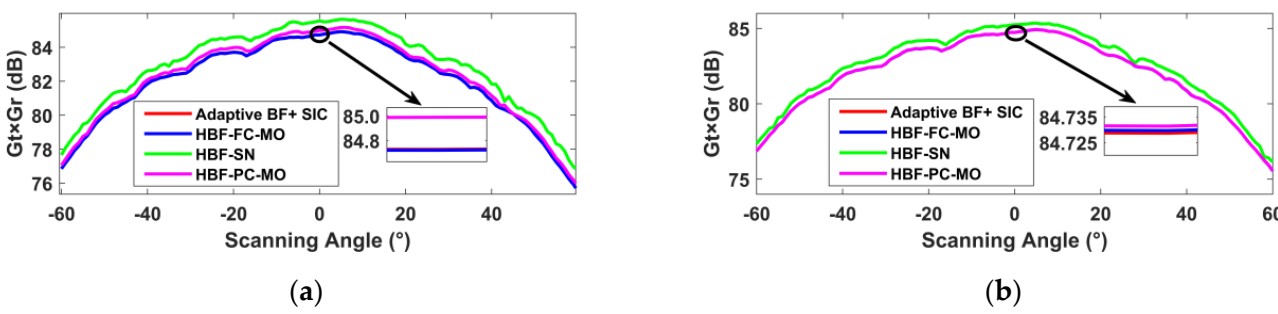

**Figure 10.** The system gain $G_t \times G_r$ of three categories of simplified structure and FD-ALSTAR (**a**) $NRF = 2$; (**b**) $NRF = 24$.

## 4. Conclusions

The HBF-FC-ALSTAR structure can achieve a degree of EII similar to that of FD-ALSTAR on the premise of ensuring the system gain and beam pattern. However, the algorithm optimization time is long, the hardware structure is complex, and the number of analog phase

shifters required at the transmitter and receiver is $NRF \times N_T$ and $NRF \times N_R$, respectively, which will be huge for large arrays. Although the HBF-PC-ALSTAR simplified structure can greatly simplify the hardware structure of the system, its EII is the worst among the three categories of simplified structure. With the increase in the number of RF links, the improvement in EII is not obvious. For this reason, the HBF-SN-ALSTAR proposed in this paper, through adding a switching network in a PC structure, realizes the dynamic connectivity of antennas, increases the redundancy and stability of the system, and greatly improves the EII of the HBF-PC-ALSTAR simplified structure. When NRF = 16, the system EII is improved by at least 18 dB. Compared with the FC structure, the hardware structure is simple and the cost is low. The algorithm optimization is also simple and fast, which is conducive to engineering implementation.

**Author Contributions:** Conceptualization, H.Y. and X.W.; methodology, H.Y.; software, H.Y. and T.L.; validation, M.X., Y.T. and D.H.; formal analysis, H.Y. and T.L.; investigation, H.Y.; resources, H.Y. and T.L.; data curation, H.Y.; writing—original draft preparation, H.Y.; writing—review and editing, H.Y. and X.W.; visualization, H.Y.; supervision, Y.T.; project administration, X.W. All authors have read and agreed to the published version of the manuscript.

**Funding:** This research was funded by the Key Areas of R&D Projects in Guangdong Province (grant no. 2019B111101001).

**Data Availability Statement:** Not applicable.

**Conflicts of Interest:** The authors declare no conflict of interest.

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
