# Peer review of "Aperture-Level Simultaneous Transmit and Receive Simplified Structure Based on Hybrid Beamforming of Switching Network"

_electronics, doi:10.3390/electronics12030602_

Round 1
Reviewer 1 Report
1. The work under review is devoted to the development of a simplified aperture-level simultaneous transmission and reception structure based on hybrid fabric beamforming in order to reduce the number of channels and power consumption, as well as to improve reliability.
2. The authors of the paper proposed four simplified structures, which are lower in cost, as they claim, since the optimization of the algorithm is simple and fast, which contributes to engineering implementation. It would be necessary to clarify how big the gain in cost is.
3. It is not clear from the work which experiment was carried out by the authors: numerical or real. Carrying out a real experiment and describing it in the work would be useful.
4. There are many abbreviations in the work, and the decoding of some of them is not given. This makes it difficult to understand the content of the work.
5. Figures 1-5 (schemes) need a more detailed description and explanation of the symbols in these figures.
6. There are two formulas in the work under the number (21) (see lines 220-222), which, of course, needs to be corrected.
7. There are no descriptions for figures 6-10 in the work, but there are descriptions for figures 11-14 and 17-18, which are not in the work.
8. The results of mathematical modeling (section 3) should be described in more detail.
Reviewer 2 Report
Congratulations for this work. The paper is good and the results are interesting.
Please find some recommendations/ remarks about the content, but also about the format:
Abstract
a aperture-level -> an aperture-level
what EII stands for?
Introduction
is growing. the STAR -> is growing. The STAR
What ‘In a strong near-field transmitting signal environment’ means? From antenna theory, near-filed and far-field have clear significance. I suppose here is not the same and maybe it should be explained.
Although Fig 1 is from [4], the used notations should be explained, especially since they repeat in the next figures as well.
the algorithms studied successively include -> the studied algorithms successively include
System Model
four types different simplified structures are obtained -> four different types of simplified structures are obtained
in the desired direction—cany be defined as -> in the desired direction—can be defined as,
Simulation experiment
Figure 6 should include a), b), c) and d)
Fig.11-14 compares the system EII of the three categories -> this statement is not correct since the article does not include those Figure numbers.
Conclusions
It is said that HBF-SN-ALSTAR improves the EII compared to HBF-PC-ALSTAR, but there is no clear and objective comparison about the mentioned improvement of hardware structure complexity. It is obvious that PC is simpler than FC, but having also SN in discussion, a table with the hardware resources used for implementation on a FPGA (for example) would be needed. Otherwise, there are only subjective statements like ‘this hardware is simpler than that one’.
